# The Role of Ca^2+^-NFATc1 Signaling and Its Modulation on Osteoclastogenesis

**DOI:** 10.3390/ijms21103646

**Published:** 2020-05-21

**Authors:** Jung Yun Kang, Namju Kang, Yu-Mi Yang, Jeong Hee Hong, Dong Min Shin

**Affiliations:** 1Department of Oral Biology, Yonsei University College of Dentistry, Seoul 03722, Korea; hannahkang77@gmail.com (J.Y.K.); NJKANG@yuhs.ac (N.K.); ymyang@yuhs.ac (Y.-M.Y.); 2Department of Dental Hygiene, Yonsei University Wonju College of Medicine, Wonju 26426, Korea; 3Department of Physiology, College of Medicine, GAIHST, Gachon University, Incheon 21999, Korea

**Keywords:** osteoclast, calcium signaling, NFAT, transient receptor potential channels

## Abstract

The increasing of intracellular calcium concentration is a fundamental process for mediating osteoclastogenesis, which is involved in osteoclastic bone resorption. Cytosolic calcium binds to calmodulin and subsequently activates calcineurin, leading to NFATc1 activation, a master transcription factor required for osteoclast differentiation. Targeting the various activation processes in osteoclastogenesis provides various therapeutic strategies for bone loss. Diverse compounds that modulate calcium signaling have been applied to regulate osteoclast differentiation and, subsequently, attenuate bone loss. Thus, in this review, we summarized the modulation of the NFATc1 pathway through various compounds that regulate calcium signaling and the calcium influx machinery. Furthermore, we addressed the involvement of transient receptor potential channels in osteoclastogenesis.

## 1. Osteoclastogenesis in Bone Remodeling

Bone remodeling is balanced by the coordinated activities of osteoclastic resorption and osteoblastic formation [1]. Imbalanced bone remodeling leads to bone diseases including osteoporosis, periodontitis and rheumatoid arthritis, which are characterized by enhanced osteoclast activity. In other words, an excessive increase in osteoclast differentiation and bone resorption gives rise to various bone-resorptive diseases [2]. Osteoclasts are the cells responsible for bone resorption. These large multinucleated cells originate from the monocyte/macrophage hematopoietic lineage [3,4]. Osteoclast differentiation depends on two essential cytokines, receptor activator of nuclear factor-κB (NF-κB) ligand (RANKL) and macrophage colony-stimulating factor (M-CSF) [5,6,7]. M-CSF is involved in the proliferation and survival of osteoclast precursors and RANKL induce osteoclast differentiation through binding to its receptor RANK and subsequent activation of nuclear factor of activated T cells (NFATc1), a master transcription factor required for osteoclast differentiation [8]. Osteoclasts are formed by the fusion of osteoclast precursor cells. Cellular fusion is an essential element in osteoclast development that results in the formation of multinucleated giant cells responsible for bone resorption activity. This process is called osteoclastogenesis. To resorb bone, osteoclasts attach to the bone surface, form a “ruffled border” and dissolve bone mineral by massive secretion of acidic elements [3].

## 2. The Role of Calcium (Ca^2+^) Signaling in Osteoclastogenesis

Ca^2+^ signaling in osteoclasts is important for multiple cellular functions, including proliferation, differentiation, gene transcription and bone resorption [9]. Ca^2+^ is released from intracellular stores, or enters the cell via plasma membrane ion channels [10]. RANKL-mediated signaling in osteoclasts is the initial step of bone resorption initiation. RANK-bound RANKL induces activation of the tumor necrosis factor (TNF) receptor-associated factor 6 (TRAF6) [11], subsequently involved in the activation of mitogen-activated protein kinases (MAPKs), nuclear factor-κB (NF-κB), and a component of activator protein-1 (AP-1) [8,12,13]. Activated NF-κB induces NFATc1 transcription to differentiate osteoclasts [6,14]. RANKL also stimulates phospholipase Cγ (PLCγ) during the early stages of osteoclastogenesis. Activated PLCγ produces inositol 1, 4, 5-triphosphate (IP_3_) in the cytosol. Son et al. [15] reported that RANKL-mediated activation of PLC induces an increase of cytosolic IP_3_ levels, which increases intracellular Ca^2+^ concentration ([Ca^2+^]_i_) through inducing its release from the endoplasmic reticulum (ER). Ca^2+^ influx through store-operated Ca^2+^ entry (SOCE) and transient receptor potential (TRP) channels causes RANKL-induced [Ca^2+^]_i_ oscillations during osteoclastogenesis [16,17,18,19]. TRP channels are involved in not only extracellular but also intracellular Ca^2+^ balance in osteoclasts [20]. Ca^2+^ release and reuptake into the ER stores is also necessary for [Ca^2+^]_i_ oscillations. Sarco/endoplasmic reticulum Ca^2+^-ATPase (SERCA) transports Ca^2+^ from the cytosol into the ER, and SERCA activity is also essential for [Ca^2+^]_i_ oscillations. Disruption of SERCA2 expression impairs RANKL-induced [Ca^2+^]_i_ oscillations [21]. Furthermore, RANKL induces a reactive oxygen species (ROS) pathway and causes long lasting [Ca^2+^]_i_ oscillations [22]. Cytosolic Ca^2+^ binds to calmodulin (CaM), which results in the activation of CaM-dependent enzymes such as the phosphatase calcineurin [23]. Activated calcineurin dephosphorylates serine residues in NFATc1, resulting in translocation of NFATc1 into the nucleus [24,25]. A recent study using Homer2 and Homer3 (Homer2/3) double-knockout (DKO) mice showed that Homer proteins regulate NFATc1 function through interaction with calcineurin to regulate RANKL-induced osteoclastogenesis [26]. Thus, increased [Ca^2+^]_i_ is a fundamental process mediating osteoclastogenesis (Figure 1). In this review, we focused on modulation of Ca^2+^ signaling through Ca^2+^ influx via TRP channels and highlighted the diverse compounds, involved in the Ca^2+^ -mediated signaling pathway, in osteoclastogenesis.

## 3. Transient Receptor Potential (TRP) Channels in Osteoclast

Cytosolic Ca^2+^ modulation is crucial in osteoclastogenesis. TRP channels are widely expressed in several mammalian tissues and involved in diverse physiological processes such as differentiation, proliferation, and apoptosis [27,28]. Several studies have focused on TRP channels as Ca^2+^-influx channels in RANKL-induced osteoclastogenesis. Generally, TRP channels are non-selective cation channels and are divided into six subfamilies: canonical (TRPCs), vanilloid (TRPVs), melastatin (TRPMs), mucolipin (TRPMLs), polycystins (TRPPs), and ankyrin (TRPA) [29]. Among the TRP channels, TRPV2 [30], TRPV4 [31], and TRPV5 [32] contribute to intracellular Ca^2+^ signaling in osteoclast differentiation. TRPC1 also regulates osteoclast differentiation through SOCE [33]. This section discusses the roles of TRPC, TRPV, and TRPML channels in osteoclastogenesis.

### 3.1. TRPC

Mildly enhanced bone mass was observed in TRPC1 null mice and its effect was revealed only in mice lacking inhibitor of MyoD family isoform a (I-mfa) [33]. TRPC1 binds I-mfa [34]. Trpc1 and I-mfa functionally interact to regulate the early differentiation stage of the osteoclast through antagonistic regulation of SOCE. Although there are limited studies on TRPC, the modulation of the Ca^2+^ release-activated Ca^2+^ current (I_CRAC_) by TRPC1, and I-mfa is crucial for NFATc1 activation and subsequent osteoclast differentiation [33].

### 3.2. TRPV

TRPV family members act as sensory channels for receptor-operated Ca^2+^ influx and are critically involved in the regulating of osteoclast differentiation [20]. The TRPV family consists of six members, TRPV1–TRPV6, composed of six transmembrane domains that form a cation-permeable pore [35,36,37].

Among the TRPV family members, TRPV1 is a non-selective cation channel activated by various stimuli such as heat, noxious stimuli, low pH, and numerous chemicals [38]. The physiological role of TRPV1 in bone biology was addressed one decade ago. TRPV1 is expressed in osteoclasts and promotes their differentiation [39]. Human osteoclast expresses functional TRPV1, as well as the cannabinoid receptors type 1 and 2 (CB1/CB2). The involvement of both receptors is controversial. Expression levels of TRPV1 are enhanced in osteoclasts derived from osteoporotic subjects, whereas CB2 are reduced [40]. More recently, TRPV1 desensitization and/or CB2 stimulation were found beneficial for reducing osteoclast over-activity [41,42]. There are several reports showing that application of the TRPV1 agonist capsaicin suppresses LPS-induced prostaglandin E2 (PGE2) production in osteoblasts and suppressed LPS-induced osteoclast formation [39,43]. On the other hand, the TRPV1 antagonist capsazepine inhibits bone formation and bone resorption activity of osteoclasts in OVX mice [44]. [6]-Gingerol, a major constituent of ginger, augments osteoclast function via TRPV1 and induces bone loss in adult ovary-intact mice [45]. Zoledronic acid is nitrogen containing bisphosphonate that inhibit bone resorption. Effects of the Zoledronic acid were antagonized by capsazepine supporting the involvement of TRPV1 channel in osteoblastogenesis and mineralization, but this mechanism is not effective in osteoclasts lacking the TRPV1 [46]. Sirtuin 1 (SIRT1), also known as nicotinamide adenine dinucleotide (NAD^+^)-dependent lysine deacetylase, directly inhibits the osteoclast differentiation by inhibiting ROS generation and TNF-α-mediated TRPV1 channel activation [47]. In addition, TRPV1, as a pain receptor, is expressed in peripheral sensory nerves [48,49]. A pathological role of TRPV1 has been revealed in both osteoporosis and osteoarthritis [41,50].

TRPV2 is closely related to TRPV1 [38,51]. TRPV2 is expressed in RANKL-treated RAW264.7 cells and TRPV2-mediated spontaneous [Ca^2+^]_i_ oscillations activate NFATc1 and promote osteoclast differentiation [30]. More recently, TRPV2 was found to regulate RANKL-dependent osteoclastic differentiation through the Ca^2+^-calcineurin-NFATc1 signaling pathway in multiple myeloma (MM) patients [52].

TRPV4 also plays an essential role in osteoclast differentiation [31]. It is known as a mechano- and osmo-sensor [53,54], and localizes to the basolateral membranes of mature osteoclasts [31]. TRPV4-mediated Ca^2+^ influx and intracellular Ca^2+^ signaling activate NFATc1 and induce osteoclast differentiation and resorption activity [31,55]. A protein–protein interaction between TRPV4 and myosin IIa regulates Ca^2+^/CaM signaling, which supports the migration and fusion of osteoclast precursors [55]. In addition, the TRPV4-specific antagonist, RN1734, inhibits osteoclast formation, whereas the TRPV4-specific agonist 4-α-PDD enhances osteoclast formation under mild acidic conditions [56,57]. Stromal interaction molecule 1 (STIM1)-mediated SOCE is involved in fluid shear stress (FSS)-induced [Ca^2+^]_i_ oscillations at the early differentiation stage of osteoclasts, whereas TRPV4 is highly associated with the Ca^2+^ response at the late stage of differentiation under FSS simulation [58]. TRPV4 knockdown significantly suppresses osteoclast differentiation and osteoporosis by inhibiting the Ca^2+^-calcineurin-NFATc1 pathway [59].

TRPV5, a highly selective Ca^2+^ channel, is activated by low [Ca^2+^]_i_ [60]. It is predominantly located on the ruffled borders of the membranes of resorbing osteoclasts [32]. TRPV5 knockout mice showed increased osteoclast numbers and reduced trabecular and cortical bone thickness [61]. In contrast, TRPV5 knockout mice had impaired osteoclastic function in vivo [32]. Although controversial, these findings suggest that TRPV5 plays an important role in osteoclastic function, again demonstrating the significance of Ca^2+^ influx in mature osteoclasts. In addition, small interfering RNA (siRNA) knockdown of TRPV5 completely inhibits RANKL-induced Ca^2+^ influx at the late differentiation stage of osteoclasts in vitro and enhances bone resorption activity in human osteoclasts [20,62]. The lack of estrogen leads to osteoporosis. Estrogen inhibits osteoclast differentiation and bone resorption activity by increasing TRPV5 expression in postmenopausal osteoporosis [63]. Song et al. also demonstrated that estrogen increases TRPV5 expression through the interaction of the estrogen receptor α (ERα) in RAW 264.7 cells. Furthermore, NF-κB binds to the putative site on the *trpv5* promoter, and TRPV5 is regulated by NF-κB [64]. Thus, TRPV5 contributes to the processes of estrogen-mediated osteoclast formation, bone resorption activity, and osteoclast apoptosis. A recent study showed that vitamin D (1,25(OH)_2_D_3_) inhibits TRPV5 expression at the early stage of osteoclastogenesis by suppressing osteoclast differentiation [65].

### 3.3. TRPML

The TRPML family has three members: TRPML1, TRPML2, and TRPML3. Among these, TRPML1 is a non-selective cation channel that permeates Ca^2+^ [66]. TRPML1 is a Ca^2+^-permeable channel in lysosomes and plays vital roles in lysosomal trafficking and functions [67]. Erkhembaatar et al. [68] showed that deleting TRPML1 inhibits RANKL-induced [Ca^2+^]_i_ oscillations, which reduces osteoclastogenesis and bone remodeling.

## 4. Diverse Compounds Modulating Ca^2+^ Signaling in Osteoclastogenesis

Osteoclasts are responsible for bone resorption and are therefore considered targets of anti-osteoporosis therapies. Novel treatment strategies aimed at preventing excessive bone resorption have been studied [69]. The study of antiresorptive agents derived from diverse compounds has become a recent topic of interest. The aim of this section is to summarize the current knowledge on diverse compounds that regulate osteoclast differentiation by modulating Ca^2+^ signaling. Thus, in this section, we mentioned by listing diverse compounds depending on their mode of action. Table 1 and Figure 2 summarize diverse compounds that regulate Ca^2+^ signaling in osteoclastogenesis. 

### 4.1. Ca^2+^-Calcineurin-NFATc1 (CCN) Pathway

#### 4.1.1. KMUP-1

KMUP-1 (7-[2-[4-(2-chlorophenyl)piperazinyl]ethyl]-1,3-dimethylxanthine), a chemical synthetic xanthine-based derivative, effectively suppresses RANKL-induced osteoclast differentiation in vitro, and also attenuated ovariectomized (OVX)-induced osteoclast differentiation and prevented bone resorption in vivo [18]. Especially KMUP-1 inhibits RANKL-induced [Ca^2+^]_i_ oscillations, and subsequently, inhibits calcineurin-NFATc1 signaling [70].

#### 4.1.2. Zinc

It has been shown that zinc, an important trace element, inhibits osteoclast differentiation by suppressing the Ca^2+^-calcineurin-NFATc1 signaling pathway in vitro and in vivo [19]. Specifically, zinc inhibits calcineurin activity but not expression and RANKL-induced [Ca^2+^]_i_ oscillations, without decreasing PLCγ phosphorylation. In addition, it was proposed that zinc inhibits calcineurin in the early stage of osteoclast differentiation and [Ca^2+^]_i_ oscillations in the middle or late stage of osteoclast differentiation [71].

#### 4.1.3. Praeruptorin A

Praeruptorin A is isolated from the dried root of *Peucedanum praeruptorum* Dunn. It also has anti-osteoclastogenic activity by inhibiting [Ca^2+^]_i_ oscillations without decreasing PLCγ phosphorylation [72].

#### 4.1.4. Cyanidin

Cyanidin, a particular type of anthocyanidins, is the sugar-free counterpart of anthocyanins. Anthocyanins are reddish pigments widely spread in colored fruits and vegetables [97,98]. Cyanidin chloride inhibits RANKL-induced osteoclast formation and osteoclast resorptive activity in vitro and protects against OVX-induced bone loss in vivo. Furthermore, cyanidin chloride impairs RANKL-induced [Ca^2+^]_i_ oscillations, which leads to the suppression of the activation of NFATc1 in cultured primary bone marrow-derived macrophages (BMMs) [73]

#### 4.1.5. Lumichrome

Lumichrome is a natural metabolite of riboflavin, a member of the B family of vitamins, and has been shown to have a beneficial effect on bone formation [99,100]. Chuan et al. [74] found that lumichrome inhibits RANKL-induced [Ca^2+^]_i_ oscillations in BMMs. Furthermore, lumichrome suppresses NFATc1, NF-κB, and MAPK signaling activation and decreases bone loss in OVX-mice by inhibiting osteoclastogenesis.

#### 4.1.6. Asiaticoside

*Asiaticoside*, a natural compound, is extracted from *Centella asiatica* and is a member of the triterpenoid family [101]. It significantly inhibits RANKL-induced [Ca^2+^]_i_ oscillations and NFATc1 expression in BMMs. Therefore, *Asiaticoside* suppresses the differentiation and function of the osteoclast via inhibiting the NF-κB and NFATc1 pathways [75].

### 4.2. PLCγ-Ca^2+^-NFATc1 (PCN) Pathway

#### 4.2.1. Oleanolic Acid Acetate

Oleanolic acid acetate (OAA) is a compound isolated from *Vigna angularis* (azuki bean). Kim et al. [76] have reported that OAA negatively regulates osteoclast differentiation by RANKL-induced PLCγ2 and [Ca^2+^]_i_ oscillations, which leads to NFATc1 activation. In vitro, OAA inhibits RANKL-induced osteoclast differentiation through PLCγ2-Ca^2+^-NFATc1 signaling. OAA administration also suppresses lipopolysaccharide (LPS)-induced bone loss in vivo.

#### 4.2.2. Harpagoside

Harpagoside (HAR), an iridoid glycoside isolated from *Harpagophytum procumbens* (devil’s claw), inhibits [Ca^2+^]_i_ oscillations via inactivation of several kinases such as Bruton’s tyrosine kinase (Btk), spleen tyrosine kinase (Syk), and PLCγ2, which leads to the suppression of RANKL-induced osteoclast differentiation [77]. HAR also restored bone density in an LPS-induced, but not in an OVX-induced bone loss mouse model in vivo [77].

#### 4.2.3. Artesunate

Artesunate is one of the effective clinical treatments for falciparum malaria [102]. It suppresses RANKL-induced Ca^2+^ influx and calcineurin expression. Furthermore, phosphorylation of PLCγ1 is decreased by artesunate treatment in RANKL-stimulated RAW264.7 cells. Therefore, artesunate suppresses RANKL-induced osteoclast differentiation and function by inhibiting the PLCγ1-Ca^2+^-calcineurin-NFATc1 pathway [78].

#### 4.2.4. Methyl Gallate

Methyl gallate (MG) is a polyphenolic compound that is known to have antioxidant [103], antitumor [104], anti-inflammatory [105], and antimicrobial activities [106]. MG is a dominant inhibitor of sodium and potassium ion channels in skeletal muscle cells [107]. Baek et al. [79] showed that MG attenuates RANKL-induced osteoclast differentiation by inhibiting both Akt (Protein kinase B) phosphorylation and intracellular Ca^2+^ influx mediated by Btk and PLCγ2.

#### 4.2.5. Berberine Hydrochloride

Berberine hydrochloride, an isoquinoline alkaloid, is found in many plants of the Berberidaceae families [108]. It inhibits the activation of PLCγ1, and thereby, inhibits Ca^2+^ influx, which reduces intracellular Ca^2+^ concentration, and subsequently, inhibits osteoclast differentiation and bone destruction through suppression of the TRAF6-Ca^2+^-calcineurin-NFATc1 signaling pathway in LPS-stimulated RAW264.7 cells [80].

#### 4.2.6. Tatarinan N

Tatarinan N (TN), a lignin-like component, is extracted from *Acorus tatarinowii Schott* [109]. It attenuates RANKL-induced osteoclast differentiation via reducing NFATc1 and c-Fos expression as well as inhibiting the ERK1/2 or p38 signaling pathway. Besides, TN significantly reduces the elevation of intracellular Ca^2+^ concentration induced by RANKL and attenuates RANKL-induced phosphorylation of Btk and PLCγ2 in a dose-dependent manner in BMMs [81].

#### 4.2.7. Physalin D

Physalin D is isolated from *Physalis alkekengi* L., known as “winter cherry”, and grows in western Asia and Europe [110]. Physalin D has been shown to have anti-inflammatory, antimalarial, and antinociceptive effects [110,111,112]. Physalin D attenuates RANKL-induced [Ca^2+^]_i_ oscillations by inhibiting phosphorylation of PLCγ2 and blocks the downstream activation of Ca^2+^/calmodulin-dependent protein kinase (CaMK) type IV and cAMP-responsive element-binding protein (CREB) in BMMs. Moreover, physalin D protects RANKL-induced bone loss in vivo [82].

### 4.3. Negative Regulation on Ca^2+^ Signaling

#### 4.3.1. Glechoma Hederacea

*Glechoma hederacea* (GH), known as ‘ground ivy’ or ‘creeping Charlie’, is a perennial hairy herb of the mint family Lamiaceae. Hwang et al. [83] have shown that GH induces a transient and large increase in [Ca^2+^]_i_, through the involvement of Ca^2+^ influx via voltage-gated Ca^2+^ channels (VGCCs), resulting in the abrogation of RANKL-induced [Ca^2+^]_i_ oscillations and the inhibition of NFATc1 expression in BMMs. However, GH-induced intracellular [Ca^2+^]_i_ elevation was independent of Ca^2+^ release from intracellular Ca^2+^ stores in BMMs. Taken together, these findings suggest that GH abrogates RANKL-induced [Ca^2+^]_i_ oscillations, inhibits NFATc1 expression, and reduces osteoclast differentiation by inactivating VGCCs.

#### 4.3.2. Portulaca Oleracea

*Portulaca oleracea* (PO), also known as verdolaga, red root, or pursley, has been widely used as traditional medicine. PO ethanol extract (POEE) has dual and contrary effects on RANKL-induced osteoclast differentiation. The POEE inhibits RANKL-induced [Ca^2+^]_i_ oscillations and NFATc1 activation, while it enhances RANKL-induced osteoclast differentiation by reducing RANKL-mediated cytotoxicity. Erkhembaatar et al. [84] proposed that RANKL-mediated cytotoxicity due to Ca^2+^ release from intracellular Ca^2+^ stores is attenuated by POEE, which leads to enhanced RANKL-induced osteoclast differentiation.

#### 4.3.3. Methotrexate

Methotrexate (MTX) is used to treat sarcoma, leukemia, and auto-inflammatory diseases such as rheumatoid arthritis [113,114]. MTX inhibits osteoclast differentiation by inhibiting RANKL-induced Ca^2+^ influx in osteoclast progenitor cells [85].

#### 4.3.4. Xanthotoxin

Xanthotoxin (XAT) is isolated from the seeds of a plant of the carrot family *Ammi majus* [115]. XAT has been shown to have antitumor activity and antioxidant activity [116,117]. Interestingly, XAT affects the intracellular Ca^2+^ levels in melanocytes, resulting in reorganization of actin stress fiber cytoskeleton [118]. Dou et al. [86] showed that XAT suppresses RANKL-induced [Ca^2+^]_i_ oscillations and the activation of downstream targets of Ca^2+^-CaMKK (Calmodulin-dependent protein kinase kinase)/Pyk2 (Proline-rich tyrosine kinase 2) signaling during osteoclast differentiation, resulting in the inhibition of NFATc1 and c-FOS in BMMs. In addition, an in vivo study showed that XAT treatment prevents bone loss and increases new bone formation in OVX-mice.

#### 4.3.5. Sinomenine

Sinomenine (SIN) is an alkaloid found in the roots and stems of *Sinomenium acutum*. SIN has been used for the treatment of rheumatoid arthritis (RA) in China [119]. SIN dramatically reduces LPS-induced upregulation of intracellular Ca^2+^ in matured RAW264.7 cells. In addition, SIN decreases expression of osteoclast-specific genes and tumor necrosis factor-α (TNF-α) production, and inhibits LPS-induced osteolysis and osteoclast differentiation in vitro and in vivo [87].

#### 4.3.6. Dried Plum Fractions

In preclinical trials, bone resorption is decreased by dietary supplementation with dried plum in ovariectomized rat and mouse models [120,121]. Graef et al. [88] showed that polyphenolic compounds in dried plums suppress intracellular Ca^2+^ signaling and MAPK signaling, resulting in the inhibition of NFATc1 expression, which reduces osteoclast differentiation in BMMs.

#### 4.3.7. KN93

KN93 is an inhibitor of multifunctional Ca^2+^/CaMKs [122]. It inhibits the formation and activation of the osteoclast. KN93 also downregulates the expression of NFATc1 and AP-1 protein family members in RANKL-stimulated RAW 264.7 cells. Furthermore, KN93 significantly decreases intracellular Ca^2+^ concentration in differentiated osteoclasts [89].

#### 4.3.8. Cajaninstilbene Acid

Cajaninstilbene acid (CSA) is a bioactive compound derived from pigeon pea leaves [123]. It suppresses osteoclast differentiation and bone resorption via inhibiting RANKL-induced ROS activity and [Ca^2+^]_i_ oscillations in RAW264.7 cells and BMMs. CSA also protects the bone loss of OVX- induced C57BL/6 mice [90].

#### 4.3.9. Methylglyoxal

Methylglyoxal is derived from organic compounds and is a precursor of advanced glycation end products. Its formation involves several metabolic pathways [124]. The formation of Methylglyoxal is increased in diabetic patients [125]. Diabetes can give rise to a state of low bone turnover osteoporosis [126]. The Methylglyoxal decreases [Ca^2+^]_I_, mitochondrial biogenesis, mitochondrial membrane potential, and glyoxalase I, resulting in the inhibition of RANKL-induced osteoclast differentiation and bone resorbing activity in RAW264.7 cells [91].

#### 4.3.10. Apocynin

The catechol apocynin (APO) is used as a NADPH oxidase (NOX) inhibitor [127]. Soares et al. [92] evaluated the effects of APO on osteoclast differentiation. APO reduces [Ca^2+^]_i_ by blocking Ca^2+^ channels except two pore segment channel 2 (TPC2) and inositol 1,4,5-triphosphate receptor type 1 (IP_3_R1). TPC2 is a Ca^2+^-permeable channel expressed in lysosomes, and IP_3_R1 is a Ca^2+^ channel that mediates Ca^2+^ release from the ER, following IP_3_ stimulation. APO inhibits osteoclast differentiation by decreasing [Ca^2+^]_i_.

#### 4.3.11. Loureirin B

Loureirin B (LrB) is an active component isolated from *Sanguis draxonis*, which is a Chinese traditional herb also known as Dragon’s Blood [128]. Yuhao et al. [93] investigated the effects of LrB on RANKL-induced osteoclast activity in vitro and in an OVX-induced osteoporosis mouse model in vivo. LrB attenuates RANKL-induced [Ca^2+^]_i_ oscillations, ROS production, and NFATc1 translocation into the nucleus in BMMs. Therefore, LrB can inhibit osteoclast differentiation and function by suppressing [Ca^2+^]_i_ oscillations, ROS, and NFATc1 activities. LrB also exerts a protective effect on OVX-induced osteoporosis in a mouse model [93].

#### 4.3.12. Calreticulin

Calreticulin (CRT) is a Ca^2+^-binding protein that regulates intracellular Ca^2+^ homeostasis by modulating cytoplasmic and ER Ca^2+^ levels [129,130,131]. Fischer et al. [94] found that exogenous CRT has an anti-osteoclastogenic effect in vitro and in vivo. Recombinant CRT Inhibits RANKL-induced [Ca^2+^]_i_ oscillations, but not ionomycin-induced Ca^2+^ influx in BMMs. Recombinant CRT also blocks expression of NFATc1 and c-Fos, but not CREB and NF-κB in RAW264.7 cells.

#### 4.3.13. 6-Shogaol

Shogaols are significant biomarkers used for the quality control of ginger-containing products and responsible for the pungent flavor in dried ginger. Among them, 6-shogaol is the most common type [132]. The 6-shogaol inhibits RANKL-induced [Ca^2+^]_i_ oscillations, ROS production, and NFATc1 activities in BMMs. Furthermore, 6-shogaol attenuates osteoclastogenesis and alveolar bone resorption in a ligature-induced periodontitis model in vivo [95].

### 4.4. Increasing [Ca^2+^]_i_ Oscillations

#### Amyloid Beta Peptide

Amyloid beta peptide (Aβ) is the principal component of the accumulations of β-amyloid found in the brains of Alzheimer’s patients [133]. Various studies have addressed the role of Aβ in osteoclasts [134,135,136]. Specifically, a recent study showed that Aβ enhances RANKL-induced osteoclast activation and functions through nuclear factor-κB inhibitor α (IκB-α) degradation, extracellular-signal-regulated kinase (ERK) phosphorylation, and increased [Ca^2+^]_i_ oscillations in BMMs [96].

## 5. Closing Remarks and Perspectives

The crucial studies on Ca^2+^ signaling in osteoclastogenesis have highlighted its role in bone biology. Considering the involvement of Ca^2+^ signaling in bone biology, the relatively few studies available to date suggest the importance of TRP channels for modulating osteoclastogenesis and bone loss. Therefore, most of the therapeutic potentials remain open. We estimate that pharmacological targeting of this membrane channels may result in the development of therapeutics that facilitate or inhibit Ca^2+^ influx.

## Figures and Tables

**Figure 1 ijms-21-03646-f001:**
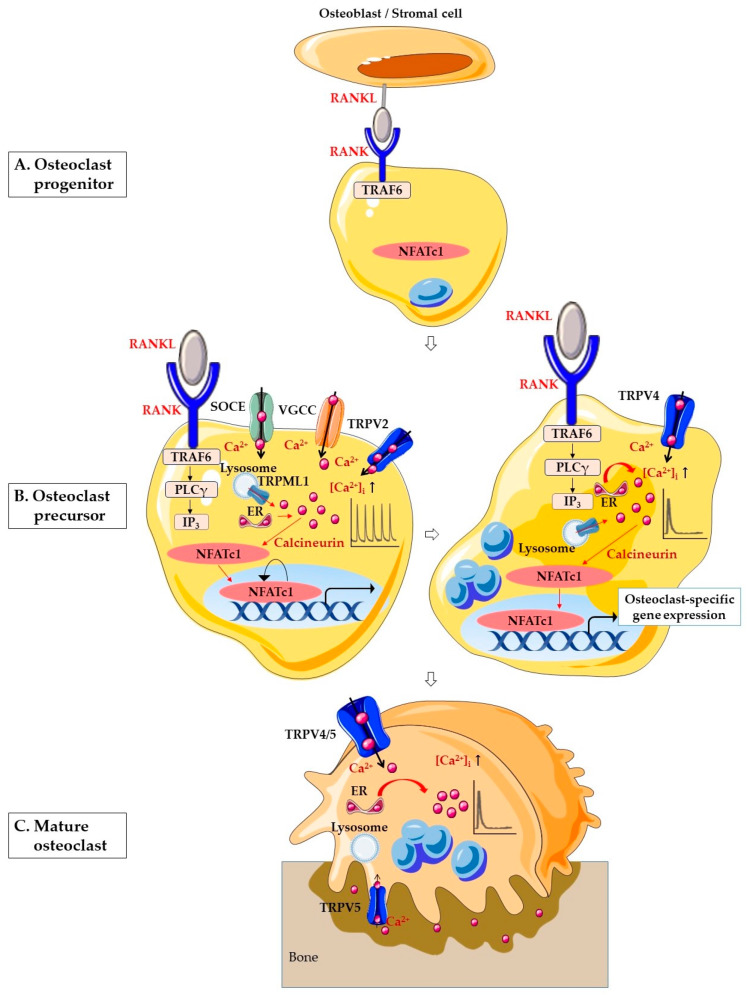
Schematic illustration of Ca^2+^ signaling in osteoclastogenesis. (**A**) RANK on the surface of osteoclast progenitor activates signaling by RANKL on the surface of osteoblasts/stromal cell to promote osteoclastogenesis. (**B**) Osteoclast precursor stage. In the early stages of osteoclastogenesis, RANK-bound RANKL induces activation of TRAF6 and stimulates PLCγ. PLCγ produces IP_3_, which evokes Ca^2+^ release from the ER. In addition, RANK-bound RANKL induces lysosomal Ca^2+^ release through TRPML1 and generates Ca^2+^ oscillation. SOCE, VGCC and TRPV2 are also involved in Ca^2+^ oscillation. The Ca^2+^ oscillations induce Ca^2+^-calcineurin-NFATc1 signaling. In the late stages of osteoclastogenesis, the Ca^2+^ oscillation is sustained by TRPV4-mediated Ca^2+^ influx. In the nucleus, NFATc1 induces the expression of various osteoclast-specific genes. (**C**) In mature osteoclasts, TRPV4 and TRPV5 in the basolateral membrane are necessary for the regulation of osteoclastic bone resorption. TRPV5 is predominantly located on the ruffled border of resorbing osteoclasts. Abbreviations: RANKL, receptor activator of nuclear factor-κB (NF-κB) ligand; RANK, receptor activator of nuclear factor-κB (NF-κB); NFATc1, nuclear factor of activated T cells cytoplasmic 1; TRAF6, tumor necrosis factor (TNF) receptor-associated factor 6; PLCγ, phospholipase Cγ; IP_3_, inositol 1,4,5-triphosphate; ER, endoplasmic reticulum; Ca^2+^, calcium; [Ca^2+^]_i_, intracellular Ca^2+^ concentration; SOCE, store-operated Ca^2+^ entry; VGCC, voltage-gated Ca^2+^ channel; TRPV2, transient receptor potential vanilloid 2; TRPV4, transient receptor potential vanilloid 4; TRPV5, transient receptor potential vanilloid 5; TRPML1, transient receptor potential mucolipin 1.

**Figure 2 ijms-21-03646-f002:**
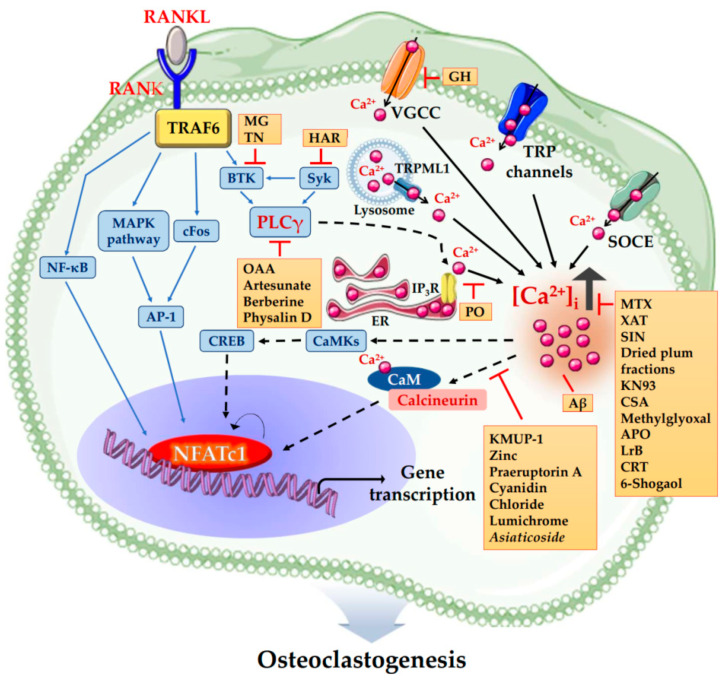
The schematic illustration summarized diverse compounds that regulate Ca^2+^ signaling in osteoclastogenesis. KMUP-1 (7-[2-[4-(2-chlorophenyl)piperazinyl]ethyl]-1,3-dimethylxanthine), Zinc, Praeruptorin A, Cyanidin Chloride, Lumichrome and *Asiaticoside* inhibit osteoclastogenesis via inhibiting Ca^2+^-Calcineurin-NFATc1 signaling independent of PLCγ. Methotrexate (MTX), Xanthotoxin (XAT), Sinomenine (SIN), Dried plum fractions, KN93, Cajaninstilbene acid (CSA), Methylglyoxal, Apocynin (APO), Loureirin B (LrB), Calreticulin (CRT) and 6-Shogaol inhibit osteoclastogenesis via decreasing [Ca^2+^]_i_. On the contrary, Amyloid beta peptide (Aβ) enhances osteoclastic bone resorption by increasing [Ca^2+^]_i_ oscillations, resulting in upregulation of NFATc1. *Portulaca oleracea* (PO) inhibits osteoclastogenesis by inhibiting Ca^2+^ release from intracellular Ca^2+^ stores. Oleanolic acid acetate (OAA), Artesunate, Berberine and Physalin D inhibit osteoclastogenesis via inhibiting PLCγ-Ca^2+^-NFATc1 signaling. Harpagoside (HAR) inhibits osteoclastogenesis via inhibiting Syk-Btk-PLCγ-Ca^2+^ Signaling. Methyl gallate (MG) and Tatarinan N (TN) inhibit osteoclastogenesis by suppression of Btk-PLCγ cascade. *Glechoma hederacea* (GH) inhibits osteoclastogenesis by inactivating VGCCs independent of Ca^2+^ release from intracellular Ca^2+^ stores. Abbreviations: RANKL, receptor activator of nuclear factor-κB (NF-κB) ligand; RANK, receptor activator of nuclear factor-κB (NF-κB); NFATc1, nuclear factor of activated T cells cytoplasmic 1; TRAF6, tumor necrosis factor (TNF) receptor-associated factor 6; MAPK, mitogen-activated protein kinases; AP-1, activator protein-1; Btk, Bruton’s tyrosine kinase; Syk, spleen tyrosine kinase; PLCγ, phospholipase Cγ; IP3R, inositol 1,4,5-triphosphate receptor; ER, endoplasmic reticulum; Ca^2+^, calcium; [Ca^2+^]_i,_ intracellular Ca^2+^ concentration; SOCE, store-operated Ca^2+^ entry; VGCC, voltage-gated Ca^2+^ channel; TRP channels, transient receptor potential cation channels; CaMKs, Ca^2+^/calmodulin dependent protein kinases; CREB, cAMP-responsive element-binding protein; CaM, calmodulin.

**Table 1 ijms-21-03646-t001:** Diverse compounds that regulate Ca^2+^ signaling in osteoclastogenesis.

Compound	Mechanism of Inhibition of RIO ^(1)^	Species	Administered Dose		Ref
			In Vitro	In Vivo	
**Mode of action: Ca^2+^-Calcineurin-NFATc1(CCN ^(2)^) signaling**					
KMUP-1	CCN signaling independently of PLCγ	RAW264.7 cell, BALB/c mice	1–10 μM	1, 5, 10 mg/kg	[70]
Zinc	CCN signaling independently of PLCγ	RAW264.7 cell, BMMs (C57BL/6 mice)	10–100 μM	N/A ^(4)^	[71]
Praeruptorin A	Inhibition of PLCγ-independent [Ca^2+^]_i_ oscillations	BMMs (ICR mice)	10 μM	N/A	[72]
Cyanidin Chloride	Suppression of NF-κB, ERK and CCN signaling	RAW264.7 cell, BMMs (C57BL/6 mice), C57BL/6 mice	5–10 μM	5 mg/kg	[73]
Lumichrome	Suppression of NF-κB, MAPK and CCN signaling	RAW264.7 cell, BMMs (C57BL/6 mice), C57BL/6 mice	7.5–10 μM	7.5 mg/kg	[74]
*Asiaticoside*	Suppression of NF-κB and CCN signaling	RAW264.7 cell, BMMs (C57BL/6 mice)	2.5–20 μM	N/A	[75]
**Mode of action: PLCγ- Ca^2+^-NFATc1(PCN ^(3)^) signaling**					
OAA	PCN signaling	BMMs (ICR mice), ICR mice	20 μM	10 mg/kg	[76]
HAR	Syk-Btk-PLCγ- Ca^2+^ Signaling	BMMs (ICR mice), C57BL/6 mice	25–100 μM	10 mg/kg	[77]
Artesunate	PCN signaling	RAW264.7 cell, BMMs (C57BL/6 mice), C57BL/6 mice	3.125–12.5 μM	5, 30 mg/kg	[78]
MG	Akt and Btk-PLCγ- Ca^2+^ Signaling	BMMs (ICR mice), ICR mice	1–10 μM	10 mg/kg	[79]
Berberine	* Inhibition of LPS-induced osteoclastogenesis through TRAF6 and PCN signaling	RAW264.7 cell	5–20 μM	N/A	[80]
TN	Suppression of Btk-PLCγ cascade, NF-κB, MAPKs and CCN signaling	BMMs (C57BL/6 mice)	1.25–5 μM	N/A	[81]
Physalin D	Suppression of PLCγ-CaMK-CREB pathway	BMMs (C57BL/6 mice), C57BL/6 mice	5 μM	10, 100 mg/kg	[82]
**Mode of action: Negative regulation of Ca^2+^ signaling**					
GH	Abrogation of RANKL-induced [Ca^2+^]_i_ oscillations by inactivating VGCCs independently of Ca^2+^ release from intracellular Ca^2+^ stores	BMMs (C57BL/6 mice)	5–50 μg/mL	N/A	[83]
PO	Suppression of RANKL-induced [Ca^2+^]_i_ oscillations by inhibiting Ca^2+^ release from intracellular Ca^2+^ stores	murine BMMs	50 μg/mL	N/A	[84]
MTX	Decrease of RANKL-induced Ca^2+^ influx	BMMs (C57BL/6 mice)	1, 5 μM	N/A	[85]
XAT	Suppression of RANKL-induced [Ca^2+^]_i_ oscillations and Ca^2+^-CaMKK-PYK2 signaling	BMMs (C57BL/6 mice), C57BL/6 mice	0.1, 1 μM	0.5, 5 mg/kg	[86]
SIN	* Inhibition of LPS-induced osteoclastogenesis by decreasing expression of NF-κB, AP-1 and Ca^2+^-NFATc1	RAW264.7 cell, C57BL/6 mice	0.25–1 mM	25, 50, 100 mg/kg	[87]
Dried plum fractions	Suppression of MAPKs and Ca^2+^ signaling, resulting in inhibition of NFATc1	RAW264.7 cell, BMMs (C57BL/6 mice)	1, 10 μg/mL	N/A	[88]
KN93	Decreasing of [Ca^2+^]_i_	RAW264.7 cell	10 μM	N/A	[89]
CSA	Block of ROS activity and [Ca^2+^]_i_ oscillations	RAW264.7 cell, BMMs (C57BL/6 mice), C57BL/6 mice	5–10 μM	10 mg/kg	[90]
Methylglyoxal	Suppression of [Ca^2+^]_i_, mitochondrial biogenesis, mitochondrial membrane potential, and glyoxalase I	RAW264.7 cell	10–200 μM	N/A	[91]
APO	Decreasing of [Ca^2+^]_i_	BMMs (C57BL/6 mice)	1 μM	N/A	[92]
LrB	Suppression of [Ca^2+^]_i_ oscillations, ROS production, and NFATc1 translocation	RAW264.7 cell, BMMs (C57BL/6 mice), C57BL/6 mice	5–10 μM	4 mg/kg	[93]
CRT	Suppression of RANKL-induced [Ca^2+^]_i_ oscillations and expression of NFATc1 and c-Fos, independently of ionomycin-induced Ca^2+^ influx	RAW264.7 cell, BMMs (C57BL/6 mice), C57BL/6, NOD mice,	0.5–500 ng/ml	0.2 mg/kg	[94]
6-Shogaol	Suppression of [Ca^2+^]_i_ oscillations, ROS production, and NFATc1 activity	BMMs (C57BL/6 mice), C57BL/6 mice	2.5–10 μM	10 mg/kg	[95]
**Mode of action: Increasing [Ca^2+^]_i_ oscillations**					
Aβ	* Enhancement of osteoclast activation by activating NF-κB, ERK and increasing [Ca^2+^]_i_ oscillations, resulting in upregulation of NFAT-c1	BMMs (C57BL/6 mice)	1–10 μM	N/A	[96]

* Another mechanism besides RIO, Abbreviations: (1) RIO, RANKL-induced osteoclastogenesis; (2) CCN, Ca^2+^-Calcineurin-NFATc1; (3) PCN, PLCγ- Ca^2+^-NFATc1; (4) N/A, not applicable; The other abbreviations are listed in the last paragraph.

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
