# Peer review of "The Role of Ca2+-NFATc1 Signaling and Its Modulation on Osteoclastogenesis"

_ijms, 2020, doi:10.3390/ijms21103646_

Round 1

Reviewer 1 Report

The manuscript/review deals with the  physiological role of calcium signaling through the calcium 2 influx/NFAT axis in osteoclastogenesis. The NFATL signaling is a well recognized pathway in different cancer disorders. It is also a well recognized drug target. Therefore, there are several points that need to be discussed before publication.

In addition to the list of natural compounds reported by the authors, several therapeutic drugs modulate the RANKL signaling pathways other than methotrexate reported by the authors and these drugs should be reported in the manuscript and the reference list should be revised accordingly.

Thalidomide and Lenalidomide are immunomodulatory drugs, which moderately increased OPG expression while inhibiting RANKL expression, abrogated osteoclast differentiation and function, and inhibited secretion of multiple myeloma survival factors from osteoclasts and bone marrow stromal cells (Raje et al. Clin Cancer Res; 25(1) January 1, 2019).

Proteome inhibitors such as bortezomib decreased serum levels of RANKL and markers of bone resorption [TRACP-5b, C-telopeptide of collagen type 1. Similarly, carfilzomib inhibits osteoclast differentiation and function through a mechanism involving disrupted RANKL-induced NF-kB signaling (Raje et al. Clin Cancer Res; 25(1) January 1, 2019).

Several anti RANKL antibodies are under development. Among these Denosumab is available and it is  specifically developed against osteoclasts formation.

Bisphosphonates BPs are also well known drugs of therapeutic interest and are known inhibitors of osteoclastogenesis with different mechanisms. Bisphosphonates BPs are internalized by osteoclasts, where they act as potent inhibitors of farnesyl pyrophosphate synthase, a key enzyme of the mevalonate pathway, involved in bone resorption, affecting osteoclast morphology and finally inducing their apoptosis (Russell, Bisphosphonates: Mode of action and pharmacology. Pediatrics 2017, 119, S150–S162;  Savino et al., Novel bisphosphonates with antiresorptive effect in bone mineralization and osteoclastogenesis. Eur. J. Med. Chem. 2018, 158, 184–200). These drugs are also reported to enhance the RANKL/OPG gene expression ratio in osteoblasts (Koch et al., Influence of bisphosphonates on the osteoblast RANKL and OPG gene expression in vitro. Clin Oral Investig 2012;16:79–86).

Within the TRPV channel family, TRPV1 channel is also functionally involved in osteoblastogenesis and mineralization and it is an emerging target for the bisphosphonate zoledronic acid. Mineralization of osteoblast cell line MC3T3-E1 and rat and mouse bone marrow-derived osteoblasts was observed in the presence of the TRPV1 agonist capsaicin and zoledronic acid (5 × 10-8⁻10-7 M); zoledronic acid and capsaicin effects were antagonized by capsazepine supporting the involvement of TRPV1 channel in osteoblastogenesis and mineralization, but this mechanism is not operative in osteoclasts lacking the TRPV1 channel (Scala et al, Zoledronic Acid Modulation of TRPV1 Channel Currents in Osteoblast Cell Line and Native Rat and Mouse Bone Marrow-Derived Osteoblasts: Cell Proliferation and Mineralization Effect. Cancers (Basel). 2019)

Author Response

Dear reviewer and chief editor,

We appreciate the reviewers for the useful comments and careful consideration to improve the paper. As instructed, we have attempted to succinctly explain changes made in reaction to all comments. Below we address the each of the comments. The original comments are provided in black color, whereas our answers are given in blue color. We obviously have needed to quote the sources correctly and done so at the places where we had missed before. In addition, the manuscript has been edited to make appropriate information to this body of work.

Reviewer 1

The manuscript/review deals with the physiological role of calcium signaling through the calcium 2 influx/NFAT axis in osteoclastogenesis. The NFATL signaling is a well recognized pathway in different cancer disorders. It is also a well recognized drug target. Therefore, there are several points that need to be discussed before publication.

In addition to the list of natural compounds reported by the authors, several therapeutic drugs modulate the RANKL signaling pathways other than methotrexate reported by the authors and these drugs should be reported in the manuscript and the reference list should be revised accordingly.

Thalidomide and Lenalidomide are immunomodulatory drugs, which moderately increased OPG expression while inhibiting RANKL expression, abrogated osteoclast differentiation and function, and inhibited secretion of multiple myeloma survival factors from osteoclasts and bone marrow stromal cells (Raje et al. Clin Cancer Res; 25(1) January 1, 2019).

Proteome inhibitors such as bortezomib decreased serum levels of RANKL and markers of bone resorption [TRACP-5b, C-telopeptide of collagen type 1. Similarly, carfilzomib inhibits osteoclast differentiation and function through a mechanism involving disrupted RANKL-induced NF-kB signaling (Raje et al. Clin Cancer Res; 25(1) January 1, 2019).

Several anti RANKL antibodies are under development. Among these Denosumab is available and it is  specifically developed against osteoclasts formation.

Bisphosphonates BPs are also well known drugs of therapeutic interest and are known inhibitors of osteoclastogenesis with different mechanisms. Bisphosphonates BPs are internalized by osteoclasts, where they act as potent inhibitors of farnesyl pyrophosphate synthase, a key enzyme of the mevalonate pathway, involved in bone resorption, affecting osteoclast morphology and finally inducing their apoptosis (Russell, Bisphosphonates: Mode of action and pharmacology. Pediatrics 2017, 119, S150–S162;  Savino et al., Novel bisphosphonates with antiresorptive effect in bone mineralization and osteoclastogenesis. Eur. J. Med. Chem. 2018, 158, 184–200). These drugs are also reported to enhance the RANKL/OPG gene expression ratio in osteoblasts (Koch et al., Influence of bisphosphonates on the osteoblast RANKL and OPG gene expression in vitro. Clin Oral Investig 2012;16:79–86).

 - Author response: We appreciate your valuable comments. We have carefully considered reviewers’ comments and suggestions. You have raised an important point. however, we believe that would be outside the scope of our paper because this review is about the role of Ca 2+-NFATc1 signaling in osteoclastogenesis. Therefore, we focused on pathway mediated intracellular Ca 2+ in osteoclast. We also mentioned the compounds related to directly Ca 2+-NFATc1 signaling in osteoclast. However, many drugs (Thalidomide, Lenalidomide, bortezomib, carfilzomib, and BPs) that you proposed are not yet studied in directly Ca 2+ signaling in osteoclast. We are afraid that we can't include these drugs (Thalidomide, Lenalidomide, bortezomib, carfilzomib, and BPs) in this review. However we think these drugs are good targets for Ca 2+ signaling study in osteoclast. Please consider our scope of review.

Within the TRPV channel family, TRPV1 channel is also functionally involved in osteoblastogenesis and mineralization and it is an emerging target for the bisphosphonate zoledronic acid. Mineralization of osteoblast cell line MC3T3-E1 and rat and mouse bone marrow-derived osteoblasts was observed in the presence of the TRPV1 agonist capsaicin and zoledronic acid (5 × 10-8⁻10-7 M); zoledronic acid and capsaicin effects were antagonized by capsazepine supporting the involvement of TRPV1 channel in osteoblastogenesis and mineralization, but this mechanism is not operative in osteoclasts lacking the TRPV1 channel (Scala et al, Zoledronic Acid Modulation of TRPV1 Channel Currents in Osteoblast Cell Line and Native Rat and Mouse Bone Marrow-Derived Osteoblasts: Cell Proliferation and Mineralization Effect. Cancers (Basel). 2019)

Author response: We appreciate your comment. We agree with you and have added a passage about the Zoledronic Acid in TRPV1 chapter with reference as you recommended.

Reviewer 2 Report

The manuscript by Kang et al. summarizes the role of Ca2+ signaling through the Ca2+ influx/NFAT axis in osteoclastogenesis. An increase in the intracellular Ca2+ concentration is key for osteoclast differentiation. Here, NFAT, in particular NFATc1 is the predominant transcription factor mediating this response. The authors have summarized the recent literature about the modulation of the NFATc1 signaling pathway via pharmacological regulation of Ca2+ signaling. Furthermore, they emphasize the role of the TRP ion channels in controlling Ca2+ signaling during osteoclast differentiation.

Before being considered for publication, the following issues need to be addressed:

  1. The title needs to be revised - Ca2+ influx axis can be misleading.
  2. The detailed description in the schematic cells is too small to be properly read on a print out. Please enlarge the figure accordingly.
  3. Abbreviations in Table 1 are not all explained in the legend, e.g. BMMs etc. Please provide a legend with a abbreviations. In general, the text is full of abbreviations, which makes it difficult to read. The authors should at least present a section where all the abbreviations are listed and explained.
  4. In the text, the compounds should not be listed one after the other, but rather combined in different chapters depending on their mode of action.
  5. Please indicate the concentration range for the different compounds that have been used in vitro and in vivo and explain how they have been administered in vivo.
  6. The transition to the TRP chapter is not obvious. Please explain a bit more in detail why TRP channels are so important in this context and allow a smooth transition from the compound chapter to the TRP chapter.

Author Response

Dear reviewer and chief editor,

We appreciate the reviewers for the useful comments and careful consideration to improve the paper. As instructed, we have attempted to succinctly explain changes made in reaction to all comments. Below we address the each of the comments. The original comments are provided in black color, whereas our answers are given in blue color. We obviously have needed to quote the sources correctly and done so at the places where we had missed before. In addition, the manuscript has been edited to make appropriate information to this body of work.

Reviewer 2

The manuscript by Kang et al. summarizes the role of Ca2+ signaling through the Ca2+ influx/NFAT axis in osteoclastogenesis. An increase in the intracellular Ca2+ concentration is key for osteoclast differentiation. Here, NFAT, in particular NFATc1 is the predominant transcription factor mediating this response. The authors have summarized the recent literature about the modulation of the NFATc1 signaling pathway via pharmacological regulation of Ca2+ signaling. Furthermore, they emphasize the role of the TRP ion channels in controlling Ca2+ signaling during osteoclast differentiation.

Before being considered for publication, the following issues need to be addressed

  1. The title needs to be revised - Ca2+ influx axis can be misleading.

-Author response: We appreciate your valuable comment. We edited to make appropriate title.

  1. The detailed description in the schematic cells is too small to be properly read on a print out. Please enlarge the figure accordingly.

-Author response: We appreciate your comment. We changed the size and arrangement in figure 1 to be properly read on a print out.

  1. Abbreviations in Table 1 are not all explained in the legend, e.g. BMMs etc. Please provide a legend with a abbreviations. In general, the text is full of abbreviations, which makes it difficult to read. The authors should at least present a section where all the abbreviations are listed and explained.

-Author response: We appreciate your comment. We modified the main captions for table 1. And we mentioned where all the other abbreviations are listed at the table below.

  1. In the text, the compounds should not be listed one after the other, but rather combined in different chapters depending on their mode of action.

-Author response: We appreciate your comment. The section of the compounds is separated and discussed depending on their mode of action in the chapter 4.

  1. Please indicate the concentration range for the different compounds that have been used in vitro and in vivo and explain how they have been administered in vivo.

-Author response: We appreciate your comment. We agree with you that the administered concentration range for the different compounds that have been used in vitro and in vivo is especially important point. Thus, we added the administered dose used in vitro and in vivo of each compounds in table1.

  1. The transition to the TRP chapter is not obvious. Please explain a bit more in detail why TRP channels are so important in this context and allow a smooth transition from the compound chapter to the TRP chapter.

-Author response: We appreciate your comment. We changed the order of the paragraphs following ‘The role of calcium (Ca2+) signaling in osteoclastogenesis’ for smooth transition. We also added additional explanations with reference why TRP channels are so important in osteoclast at line 55.

Round 2

Reviewer 1 Report

Despite the authors did not fully agree with the reviewer comments, the manuscript was revised and improved.